# High-Frequency Ultrasonography in Hair and Nail Disorders—How It May Be Helpful

**DOI:** 10.3390/diagnostics15030332

**Published:** 2025-01-30

**Authors:** Adriana Polańska, Dominik Mikiel, Marta Szymoniak-Lipska, Barbara Olszewska, Aleksandra Dańczak-Pazdrowska

**Affiliations:** Department of Dermatology, Poznan University of Medical Sciences, 60355 Poznan, Poland; mikielki@gmail.com (D.M.); szymoniaklipska@gmail.com (M.S.-L.); b.olszewska@gumed.edu.pl (B.O.); aleksandra.danczak-pazdrowska@ump.edu.pl (A.D.-P.)

**Keywords:** HFUS, high-frequency ultrasonography, skin appendages, nail unit, cicatrical alopecia, alopecia areata, ultrasonogrpahy of the psoriatic nails

## Abstract

Ultrasonography is a recognized method of tissue visualization in medicine, which is based on the use of mechanical waves (ultrasound). Its application in dermatology requires the use of higher frequencies, hence the use of high-frequency ultrasonography (HFUS) is recommended. HFUS has gained approval in many areas of dermatology, including imaging of skin appendages [nail unit (NU) and hair follicles]. The analysis of the changing echogenicity of structures under the influence of inflammatory and neoplastic processes is used to assess the extent of the disease, treatment planning as well as in monitoring therapeutic effects. The aim of our work is to present the possibilities of visualizing NUs and scalps with the use of HFUS.

## 1. Introduction

The nail unit (NU) and hair (including hair follicles) are classified as skin appendages and, therefore, easily accessible for clinical examination. Among the many non-invasive diagnostic methods used in the assessment of skin appendages, especially those supporting value in everyday dermatological practice, dermatoscopy (onychoscopy) and its variant in relation to hairy skin and hair assessment, trichoscopy, are used [1,2]. The importance of these methods has been confirmed in diagnosing and monitoring the activity of many disease entities [3,4]. One of the imaging methods, the significance of which in dermatology has been growing in recent years and may supplement dermoscopic evaluation, is ultrasonography, related to the use of real-time ultrasonic waves. In dermatological practice, transducers emitting high-frequency waves are essential, which allows for the assessment of superficial structures. Therefore, in high-frequency ultrasonography (HFUS), the device generates a wave with a frequency of 20 MHz or higher [5,6].

Assessments with the use of HFUS are painless and free from the risk of infection or oncogenic potential resulting from ionizing radiation, and, thanks to its repeatability, it can be used at various stages of the diagnostic and therapeutic process in a wide range of skin diseases, including neoplasms and inflammatory conditions [5,6,7,8]. Ultrasound assessment is based on the analysis of changes in echogenicity, and many inflammatory and neoplastic processes are associated with a reduction in this parameter. This method allows for visualization of individual components of the NU (dorsal plate, ventral plate, matrix) and assesses the morphological changes to which these structures are subjected as a result of the inflammatory or neoplastic processes [9,10]. In the evaluation of hairy skin, HFUS allows for the observation of hair follicles, and previous studies have confirmed the possibility of their visualization using ultrasound in a wide range of frequencies in various anatomical areas of the body [11,12].

Based on a literature review and our own experience, we aim to present the current possibilities of imaging skin appendages with the use of HFUS, including the visualization of NUs and hair, in clinical practice.

## 2. Hair Follicles in HFUS

### 2.1. Healthy Hair Follicles in HFUS

Most hair follicles in the growth phase (anagen) reach the lower layers of the dermis and subcutaneous tissue. In order to fully image hair follicles, especially in the scalp, it is necessary to use transducers that use frequencies that allow for sufficiently deep penetration (including even lower frequencies like 10–15 MHz) [12,13,14]. On the other hand, differentiating individual structures associated with hair follicles, such as sebaceous glands, hair shafts, and peripilaris muscles, requires higher frequencies (30–70 MHz) [11]. The higher the resolution of the examined area, the less penetration of the examined area, contributing to the reduction of the possible visualization of hair follicles. This is especially important in areas characterized by greater skin thickness and the occurrence of long hair follicles, as in the case of the scalp. Assessment of skin with numerous, densely arranged hair follicles, especially on the scalp, requires appropriate preparation [1]. It is recommended that hair covering the examined area be removed by shaving it down to the skin level beforehand. Nonremoved hair shafts cause an acoustic shadow, which negatively affects the quality of the obtained ultrasound images and makes their interpretation difficult. Analysis of hair follicles for the presence and intensity of inflammation should be extended to include an assessment of vascular flow [12,14,15,16]. Doppler ultrasound transducers are typically used for this examination.

Healthy hair follicles are distinct, elongated hypoechoic structures arranged parallel to one another and obliquely or perpendicularly to the entrance echo [13]. The echogenicity of hair follicles is lower than the echogenicity of the surrounding dermis and similar to the echogenicity of the subcutaneous tissue band (Figure 1). The proximal end is distinct and visible under the entrance echo. In some images, a narrow, hypoechoic section can be additionally visualized, breaking through the entrance echo. This corresponds to the openings of hair follicles on the surface of the epidermis. Significant differences were demonstrated in the position of individual hair follicles depending on the phase of the hair cycle [12]. Hair follicles in the anagen phase are significantly longer and located much deeper than in the case of hair follicles in the resting phase (telogen), which translates into visualizing their distal ends. Unambiguous determination of the distal end is only possible in the case of follicles located in the dermis (difference in echogenicity). Anagen follicles, especially in the area of the scalp, located in the subcutaneous tissue, most often do not show a clear difference in echogenicity compared to the surroundings, which makes it much more difficult to determine their boundaries (Figure 1). Differences were found in the length and thickness of individual hair follicles depending on the anatomical location examined [11]. Wortsman et al. described ultrasound images of hair follicles [11]. Hair shafts were imaged in the axillae, groin, and facial skin (eyelids and eyebrows). They were described as two-layered structures with higher echogenicity than the surrounding follicle elements. In some of the examined areas, distal extension and bifurcation of the shaft base, corresponding to the hair bulb, were detected. The peripilaris muscle imaged in the skin of the trunk and extremities appeared as a hypoechoic, distinct, thin band extending obliquely from the hair follicle toward the upper dermis. Sebaceous glands were described as hyperechoic, oval structures attached to hair follicles. They were particularly prominent in the skin of the face and proximal extremities.

### 2.2. Inflammatory Diseases of the Scalp in HFUS

In addition to descriptions of normal hair follicles, ultrasound images of follicles in various pathological conditions were also studied, mainly in the scalp [17,18,19]. Available publications have proven differences in hair follicle morphology by comparing ultrasound images of healthy individuals and those with various forms of alopecia [18,19,20]. It was found that hair follicles exhibit varied echogenicity depending on the presence of hair shafts, inflammation or accompanying fibrosis. Published studies indicate that visualization of hair follicles using ultrasound in different diseases may be particularly useful in differentiating between cicatricial and non-cicatricial alopecia and in assessing the phase of alopecia areata before deciding on potential therapy [18,19]. In Table 1, we present the summary of different alopecias and their HFUS possible disturbances.

Seborrhoeic dermatitis

Sonograms of patients with seborrhoeic dermatitis were morphologically similar to those observed in healthy individuals. However, the distinguishing feature was an evident enlargement of the distal sections of the hair follicles [18].

Alopecia areata (AA)

In this entity, ultrasound images differ depending on the phase of the disease. In the active stage of AA, distinct, dilated hair follicles with a morphology resembling water drops were observed (Figure 2). In the inactive phase of AA, hair follicles with unclear boundaries, fewer in number, and reduced echogenicity of the band corresponding to the dermis were described. The regrowth phase was associated with the occurrence of structures typical of sonograms of healthy individuals with predominant follicles of regular width without a distinct distal end and, additionally, thinner follicles with distinct ends [18].

Lichen planopilaris (LPP)

The previous report demonstrated differences in ultrasound images depending on the phase of LPP [19]. Enlarged hair follicles with distinct proximal and distal segments with a cigar-shaped morphology were typical of the active phase. Reduction of skin thickness, decreased number of follicles, a saw-shaped dermal–subdermal border, and increased skin and follicle echogenicity were typical of inactive LPP (Figure 3).

Frontal fibrosing alopecia (FFA)

The ultrasound images of patients with FFA, both in the active and inactive phases, were found to resemble those observed in patients with LPP, indicating a common origin of both diseases [19]. Porrino-Bustamante ML et al. demonstrated a higher vessel diameter and flow in color Doppler ultrasound in the hairline implantation area in patients with FFA compared to the control group [15]. In another article, ultrasound images of a patient with FFA were described using a 33 MHz transducer [21]. The following features were noted: increased echogenicity of the skin, loss of differentiation between dermis and epidermis, more superficially inserted follicles (in telogen phase), and few hair follicles with an enlarged and hypoechoic area around them. Whittle et al. used Doppler ultrasound to assess the scalp of patients with frontal cicatricial alopecia before and after hair transplantation, considering it to be a potentially beneficial device for monitoring the effects of this surgical procedure [22].

Discoid lupus erythematosus (DLE)

In the active form of DLE, focal doubling of the entrance echo band, overlapping of follicular structures with a tendency to blur the boundaries, and formation of a hypoechoic band below the entrance echo were observed [19]. A significantly reduced number of hair follicles, increased echogenicity of the skin hair follicles, and reduced skin thickness were features described during the inactive stage of DLE.

Dissecting cellulitis of the scalp (DC)

Color Doppler ultrasound has proved helpful in supporting an early diagnosis of DC [16]. Dermal and hypodermal hypoechoic fluid collection connected to the base of widened hair follicles and increased vascularity in the periphery of the fluid collection was described in the case of the early stage of DC. The authors suggest that ultrasonography can be helpful in distinguishing DC from epidermal and trichilemmal cysts.


**Other scalp diseases**


Androgenetic alopecia (AGA)

Ultrasound images of patients with AGA were characterized by a reduced number of hair follicles and their heterogeneity, both in terms of width and the ability to visualize distal sections of hair follicles, which is associated with the process of miniaturization, which is typical for this disease entity [18].

Cutis vertis gyrate

Ortiz-Orellana et al. found in images of patients in the US with cutis veriticis gyrata areas with increased skin band and subcutaneous tissue thickness. However, they did not find significant deviations in the width of the examined hair follicles [23].

Lipedematous alopecia

Lobato-Berezo et al. studied the usefulness of ultrasound in patients with lipedematous alopecia of the occipital region [24]. A distinct, asymmetric increase in the thickness of the subcutaneous tissue, its partial penetration into the band corresponding to the dermis, and an indistinct boundary between the bands of the dermis and subcutaneous tissue were described.

## 3. NUs in HFUS

### 3.1. Healthy NUs in HFUS

HFUS allows the visualization of the nail plate (NP) in the form of two hyperechoic, parallel bands corresponding to the ventral and dorsal laminae. A linear hypoechoic structure (middle lamina) is visible between the laminae [9]. Proximal to the described structures is the cuticle, with echogenicity similar to the dorsal and ventral laminae, and the proximal nail fold, with echogenicity lower than the aforementioned laminae. The matrix is visible as a hypoechoic structure with blurred boundaries adjacent to the proximal nail fold and the proximal part of the ventral lamina. The matrix can be visualized as a hypoechoic structure. However, its visibility can depend on the equipment used (kind of probe) [9,25]. The NU visualization with HFUS is presented in Figure 4.

The thickness of the NP is related to the type of finger and differs depending on gender [9]. In both the dominant and non-dominant hand, the NP thickness decreased with distance from the thumb [9,25]. The average thickness of the NP ranged from 0.373 mm in the fifth fingers to 0.434 mm in the second fingers. It can be assumed that the observations are related to the greater participation of the radially located fingers in everyday activities. Greater exposure to pressure is associated with the need for more effective protection of the fingertips.

Moreover, the mentioned gender plays a role in the thickness of the NP. Our work indicates that in the group of women, the average thickness of the NP ranged from 0.347 mm in the fifth fingers of the non-dominant hand to 0.411 mm in the second fingers of the non-dominant hand. The group of men ranged from 0.399 mm in the fifth fingers of the non-dominant hand to 0.457 mm in the second fingers of the non-dominant hand [9]. Wollina et al. observed [26], that gender is related to the thickness of the matrix and is greater in the group of men. Larger dimensions of the NU components in men may be related to the action of hormones and analogous to the differences in skin thickness in men and women. In a 2012 study on a group of 88 volunteers using HFUS, men were shown to have 10–20% thicker skin than women (skin of the forehead, cheek, jaw and subumbilical region) (*p* < 0.003) [27]. Data on differences in the thickness of NU components depending on age are not clear. Szymoniak-Lipska et al. observed no statistically significant relationship between NP thickness and age in most fingers, while Wollina U. et al. indicated that in both men and women, the matrix volume increased with age [9,26].

### 3.2. Infectious and Inflammatory Diseases of the NU in HFUS

Fungal infection

Onychomycotic nails show a loss of interplate space proximally, a swollen proximal nail fold, and decreased plate/distal interphalangeal joint distance. The other features described in onychomycosis are also the thickening of the nail bed and NP (Figure 5) [28].

Psoriasis

The sonographic features of psoriatic onychopathy include thickening of the NP and nail bed, with a positive correlation with the severity of the disease expressed on the nail psoriasis severity index (NAPSI) [29,30]. Additionally, a spectrum of qualitative alterations has been described in the literature, such as the presence of hyperechoic deposits in the ventral lamina (Figure 6A) blurring of the border of the ventral lamina of the NP (Figure 6B), waviness of both laminae of the plate (Figure 6C), and in the case of severe onychopathy, a complete loss of the bilaminar structure (Figure 6D) [31]. The pathognomic sign in psoriasis is the oil spot presented in Figure 7 as a hyperechogenic heterogeneous area with blurred boundaries.

Deposits in the ventral lamina are a particularly promising ultrasound feature (Figure 6A). They were described in the literature more often in the group of patients with clinically visible onychopathy than in those without it [32]. It was also the only ultrasound finding described in patients without visible nail changes [33]. From our observations carried out on a group of 61 people with psoriasis, including 34 with onychopathy and 38 healthy people, ventral lamina deposits were observed more frequently statistically in psoriatic patients with nail involvement than in patients without nail changes (62% vs. 23%) and were not observed in healthy subjects.

Moreover, ultrasound images are associated with normalizing nail findings. Using HFUS, the intensity of local inflammation, enthesopathy, and early development of psoriatic arthritis can be measured and evaluated [25].

Lichen planus

The data in the literature on the ultrasound presentation of lichen planus of the nails are scarce. A single study described thickening and decreased echogenicity of the nail bed, a hypoechogenic area (“halo”) surrounding the proximal nail plate, thickening and decreased echogenicity of the proximal nail fold, and increased vascularity of the nail bed on color Doppler (Figure 8) [34].

Alopecia areata (AA)

To date, one case has been published describing HFUS (33 MHz) usage in evaluating nail changes in AA [35]. The patient was a 12-year-old boy with nail pitting and trachyonychia. The HFUS images showed focal nail plate thickening with loss of bilaminar structure and hyperechoic bands between the ventral and dorsal laminae. The observed features corresponded clinically to focal nail roughness [35].


**Other NU diseases**


Retronychia

Abnormal growth of the nail plate into the proximal nail fold, called retronychia, can be confirmed by HFUS. Sonographic features of retronychia include overlapping NP, a hypoechoic area (“halo”) surrounding the proximal part of the NP, a decrease in the distance between this part and the base of the distal phalanx compared to the healthy nail apparatus on the other limb, and a thickening of the proximal nail fold by 0.3 mm or more compared to the healthy limb [36,37,38].


**Benign NU tumors in HFUS**


Pyogenic granuloma (lobular capillary hemangioma)

The ultrasonographic presentation of pyogenic granuloma is a hypoechoic nodule with blurred borders [39]. On color Doppler, it is usually hypovascular. The exception is the telangiectatic variant, which shows increased vascularity with low-velocity flow vessels [37].

Glomus tumor

Most of these tumors are located in the phalanges of the hands, predominantly subungually. On HFUS, they appear as well-defined round or oval hypoechoic nodules in the nail bed, most often located just above the periosteum of the distal phalanx [40]. The lesion often models the phalanx bone, causing a scalloping of the bone margin without blurring the border (erosion) of the bone. According to A. Sechi et al., that may be important in differentiating this tumor from squamous cell carcinoma located subungually [41]. Eighty perent of glomus tumors show hypervascularization on color Doppler, while rarer hypovascular variant concerns tumors derived mainly from blood vessel cells (glomangioma) [37].

Subungual hemangioma

A characteristic feature of soft tissue hemangiomas is the presence of calcifications, which can be visualized on ultrasound and radiological examination. Sonographically, subungual hemangiomas appear as hypoechoic nodules with partially blurred borders [42]. Tumors with adipocyte components, however, can have mixed echogenicity. Calcification is visible as a strongly hyperechoic point with an area of reduced echogenicity (acoustic shadow) located distally from the transducer. On color Doppler, hemangiomas are solid tumors with increased vascularity and an atypical low-resistance arterial signal [42].

Onychomatricoma

The ultrasound image includes the presence of a homogeneous, hypoechoic lesion with blurred boundaries within the nail matrix and hyperechoic bands within the proximal part of the NU. On color Doppler examination, the degree of vascularity is variable; most often, low flow is observed at the lesion’s periphery [37].

Onychopapilloma

The HFUS image of onychopapilloma is quite characteristic. It includes the formation of an oval or linear hyperechoic lesion in the nail matrix, bulging into the nail plate and running linearly in the intraplate space, which can be well visualized by positioning the transducer across the NP. Onychopapillomas show no vascularity on color Doppler [42,43,44].

Subungual exostosis

In an ultrasound examination, exostosis is visible as a strongly hyperechoic mass connected to the bony margin of the distal phalanx. The nail bed is often thickened due to inflammation and granulation tissue around the lesion [37].

Viral wart

These lesions are visible on HFUS as hypoechoic solid nodules with a characteristic, fusiform shape. They are often located in the nail bed and the adjacent nail fold. On Power Doppler examination, they showed variable vascularity, which can be observed in the lower part of the wart. At the same time, the nail plate may have an upward protrusion and irregularities within both of its laminae [44].

Mucous cyst

On HFUS, mucous cysts are visible as hypoechoic, oval structures, in 80% of cases occurring within the proximal nail fold, pressing on the nail matrix, with an artifact in the form of acoustic enhancement behind the lesion [37] (Figure 9). In rare cases, mucous cysts may occur submatrixally -Sometimes, it is possible to locate a canal between the cyst and the joint capsule in the form of an anechoic, twisted band. During the removal of the lesion, efforts should be made to close this connection to decrease the possibility of relapse [45].


**Malignant NU tumors in HFUS**


Squamous cell carcinoma (SCC)

In the HFUS image, SCC is visible as a hypoechoic lesion in the nail bed, often with blurred borders and showing signs of invasiveness with peripheral, punctate hypoechoic areas (foci). The lesion may invade the bone, causing its erosion, which is visible as a blurring of the periosteal border [41]. In the case of large tumors, partial loss of the nail plate and its bulge may be observed. Color Doppler shows strong vascularization within the tumor and surrounding tissues [37]. A single study described artifactual changes (acoustic shadow) behind the tumor [41].

Keratoacanthoma

On ultrasound, an area with blurred boundaries with an anechoic center and hypoechoic rim is visible within the nail bed. The lesion may blur the boundaries of the ventral lamina of the nail plate, undermine it, and cause scalloping of the bony margin of the distal phalanx [37].

Melanoma

Most melanomas of the NA are in situ tumors and cannot be visualized on HFUS due to the resolution limitations of ultrasonography in detecting superphicial melanin [37]. Thicker melanomas present sonographically as homogeneous, hypoechoic, hypervascular structures with blurred margins, visible within the nail bed, sometimes destroying the nail plate or phalanx bone. HFUS potentially allows for determining the thickness of the lesion (Ultrasound Breslow Index) and can help assess the local stage of disease advancement [46]. The echogenicity of benign and malignant NU tumors is summarized in Table 2.

## 4. Conclusions

HFUS, as a non-invasive visualization method, can be a valuable supplement in diagnosing hair and nail diseases in everyday clinical practice. It allows for structural assessment and can be applied in the evaluation of the effects of therapy and, due to safety, in long-term monitoring of its effects. In the area of the scalp, it allows the assessment of hair follicles, sometimes along their entire length, and can replace biopsy in assessing disease activity. The possibility of in vivo observation of hair follicles can have an impact on therapeutic decisions. A detailed examination of NUs may show the origin of the tumor and assess its vascularity, which may be especially helpful in planning the surgical intervention. The exact diagnosis of the histopathological type of tumor of an NU is limited with HFUS; however, some features like invasion into the bones may help in differentiation. It should be mentioned that HFUS evaluation is based on the analysis of changes in echogenicity, and many inflammatory and neoplastic processes are associated with a decrease in this parameter, which may limit the specificity. Therefore, a comprehensive ultrasound evaluation should always be performed in relation to clinical aspects [47]. In future perspectives, artificial intelligence and machine learning-based methods can increase diagnostic accuracy [7]. As a painless examination method, it is safe for children and pregnant women. Unfortunately, the widespread use of ultrasonography in the assessment of skin appendages is limited by the cost of the equipment and the need for training in this field.

## Figures and Tables

**Figure 1 diagnostics-15-00332-f001:**
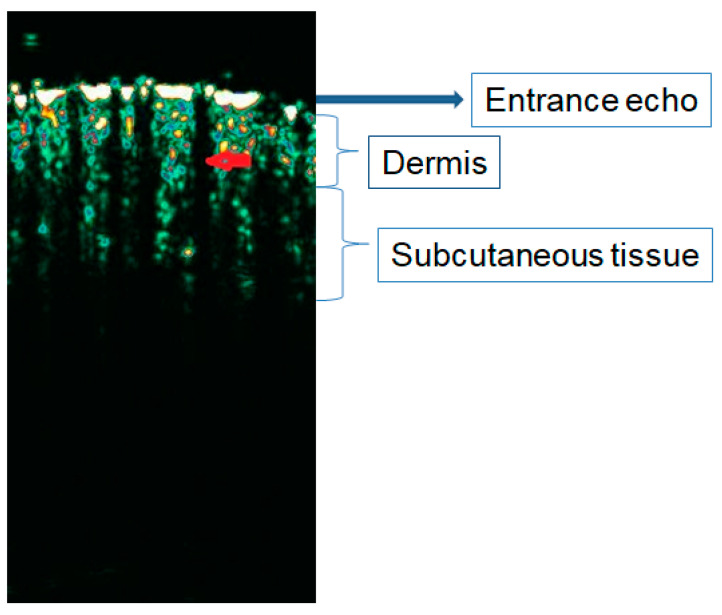
Healthy hairy skin of the scalp (frontal region) in HFUS (20 MHz). Red arrow: hair follicles in the anagen phase.

**Figure 2 diagnostics-15-00332-f002:**
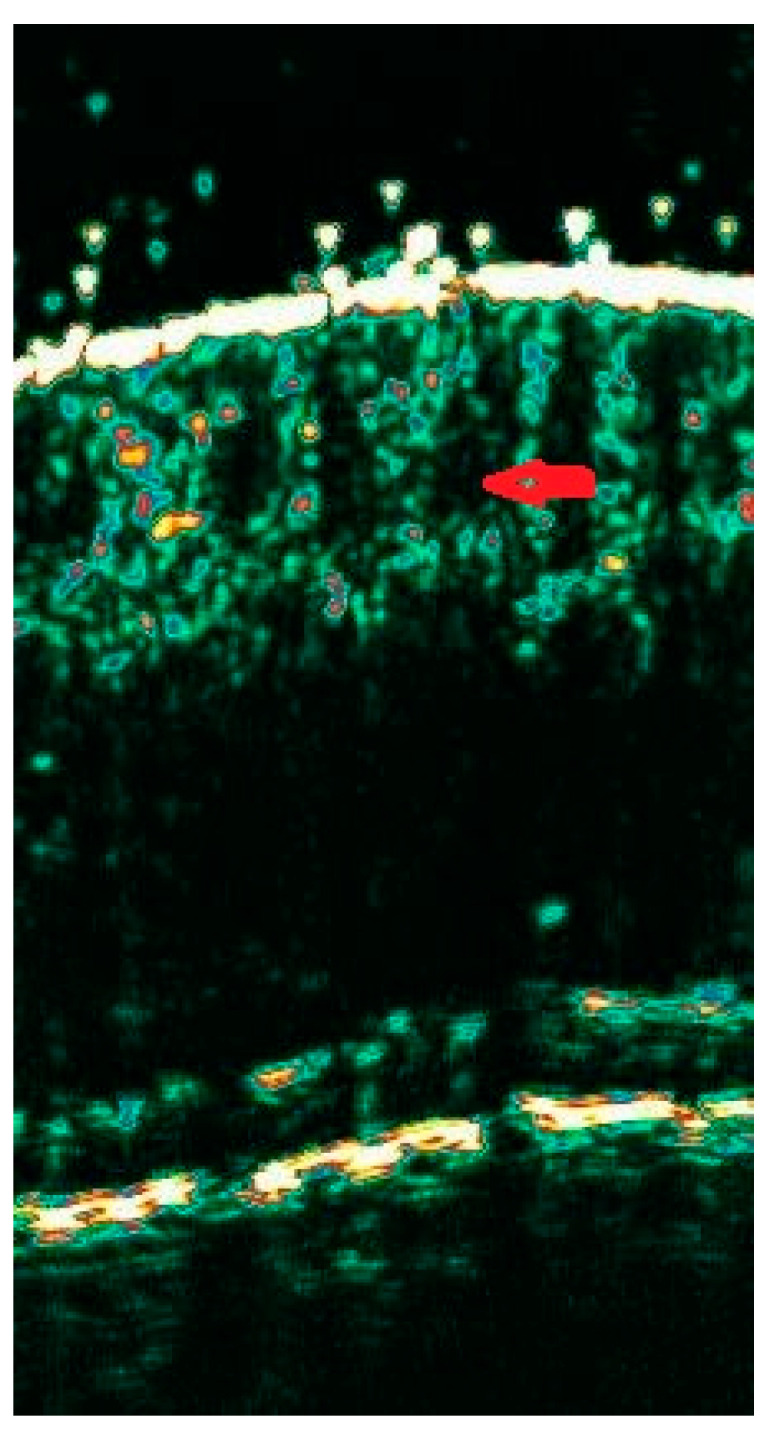
Active phase of AA—red arrow presenting dilated hair follicles mimicking water drops.

**Figure 3 diagnostics-15-00332-f003:**
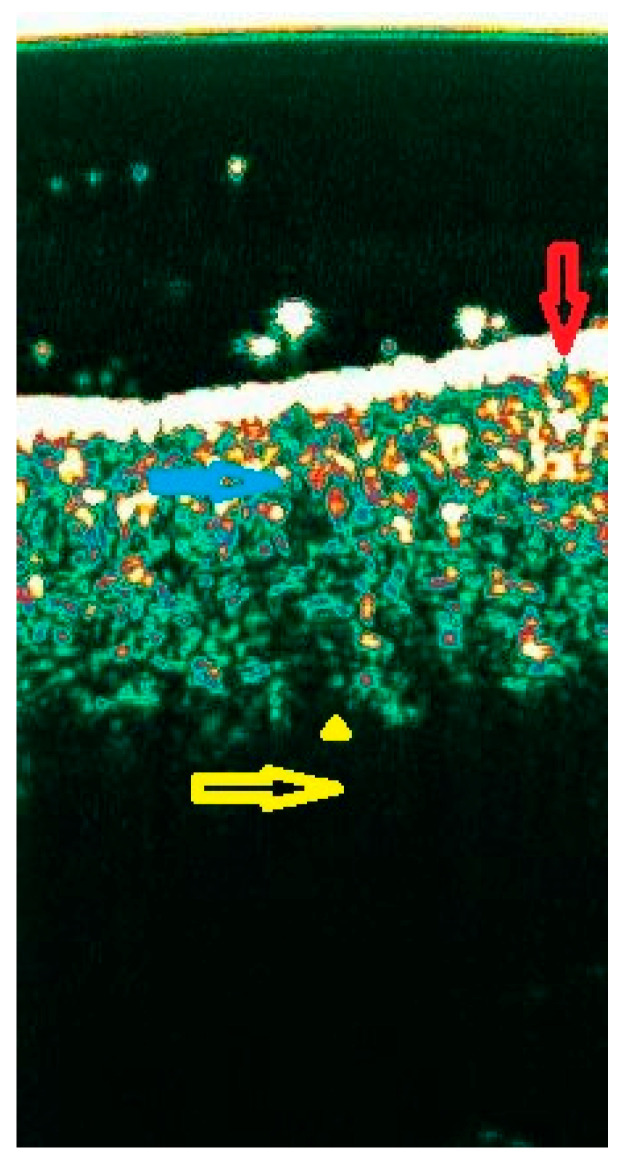
LPP in inactive phase with increased echogenicity of the dermis related to the skin fibrosis with the remaining hair follicles. Red arrow: entrance echo; blue arrow: dermis with the remnant of the hair follicle; yellow arrow: subcutaneous tissue (anechoic) with saw-shaped dermal–subdermal border (yellow triangle).

**Figure 4 diagnostics-15-00332-f004:**
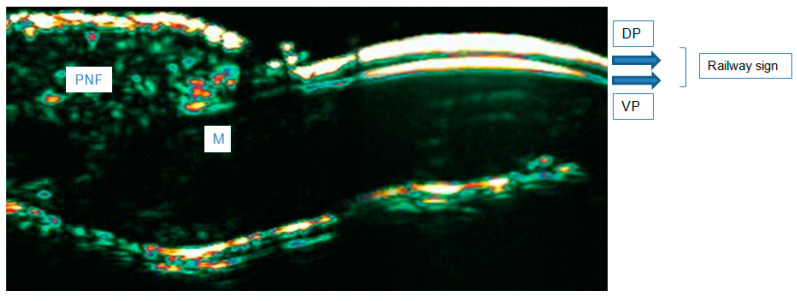
Healthy NUs in HFUS (20 MHz). Abbreviations: PNF—proximal nail fold; M—matrix, DP—dorsal plate, VP—ventral plate.

**Figure 5 diagnostics-15-00332-f005:**
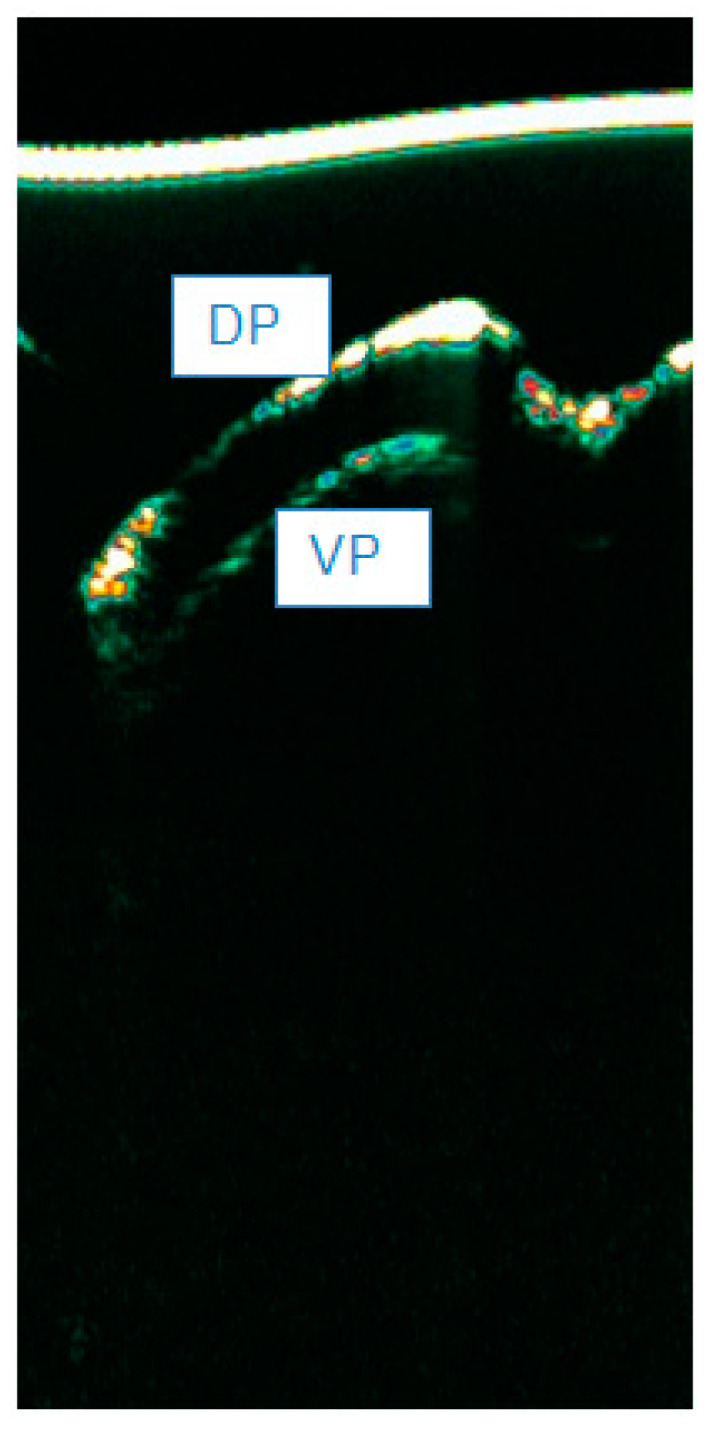
Dermatophyte infection of the NU with pronounced thickening of the NP in HFUS (20 MHz).

**Figure 6 diagnostics-15-00332-f006:**
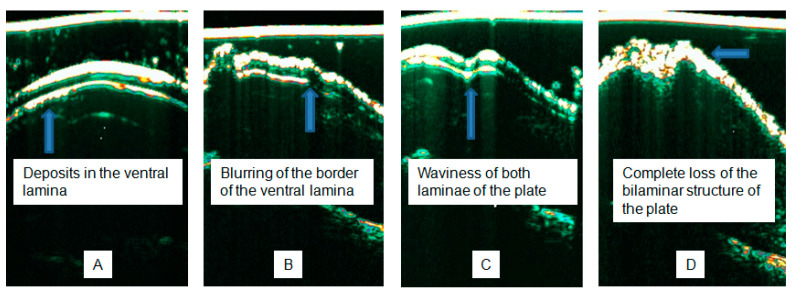
Distinct sonographic features of psoriasis, according to Wortsman (**A**–**D**) [31].

**Figure 7 diagnostics-15-00332-f007:**
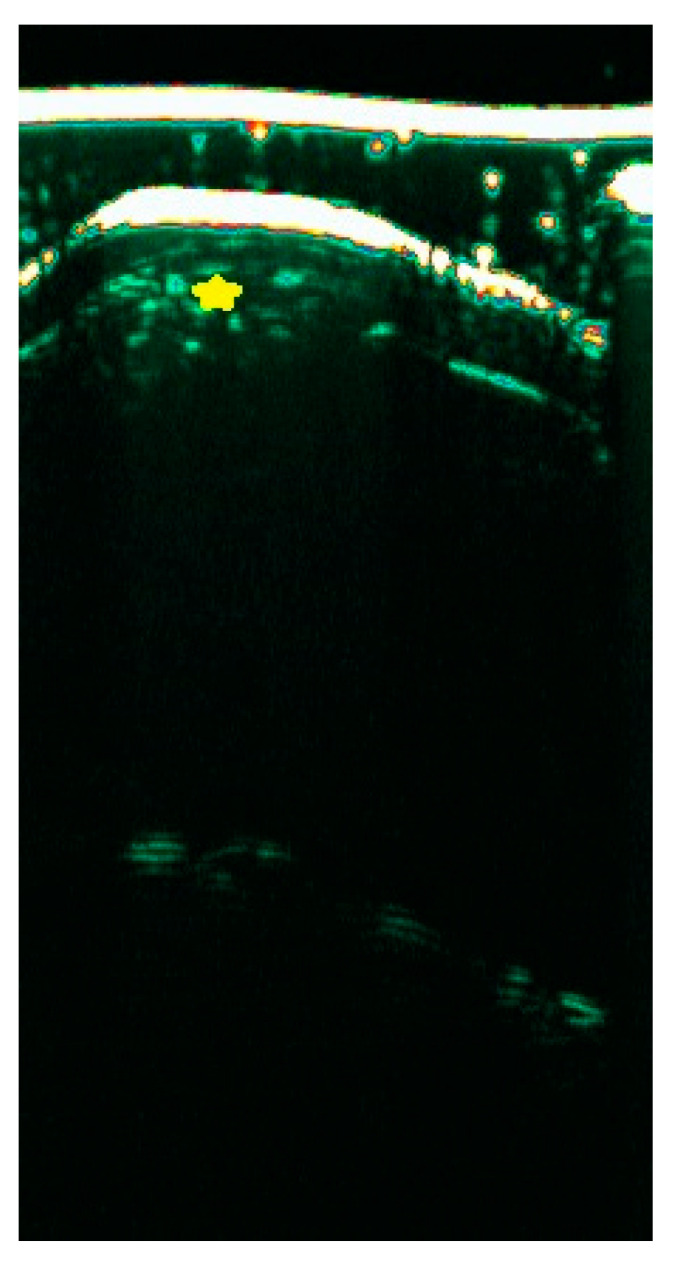
The psoriatic oil spot in HFUS (yellow star) (20 MHz).

**Figure 8 diagnostics-15-00332-f008:**
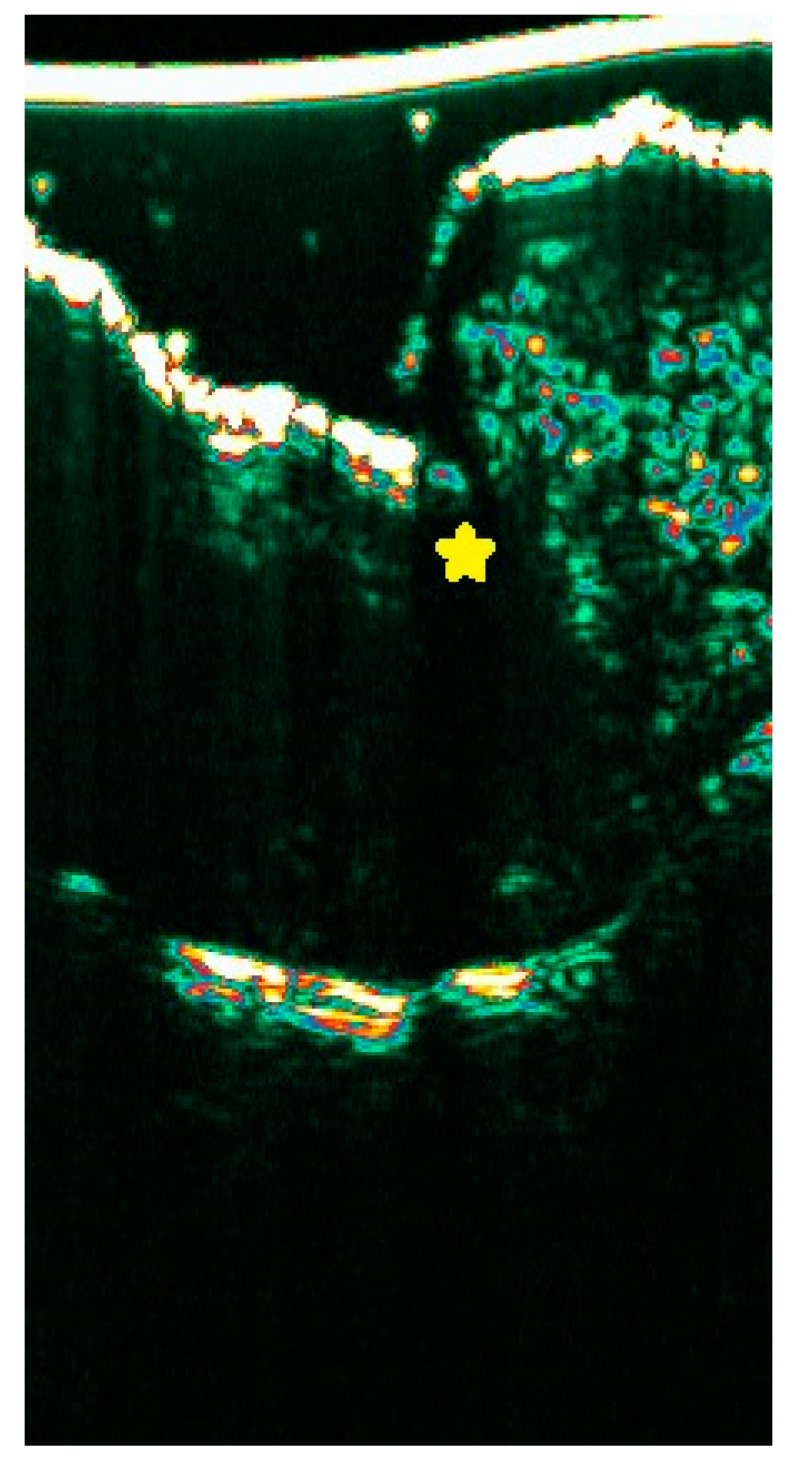
Lichen planus nail changes in HFUS (20 Mhz) with visible hypoechogenic area (“halo”) surrounding the proximal NP (yellow star) and lack of railway sign.

**Figure 9 diagnostics-15-00332-f009:**
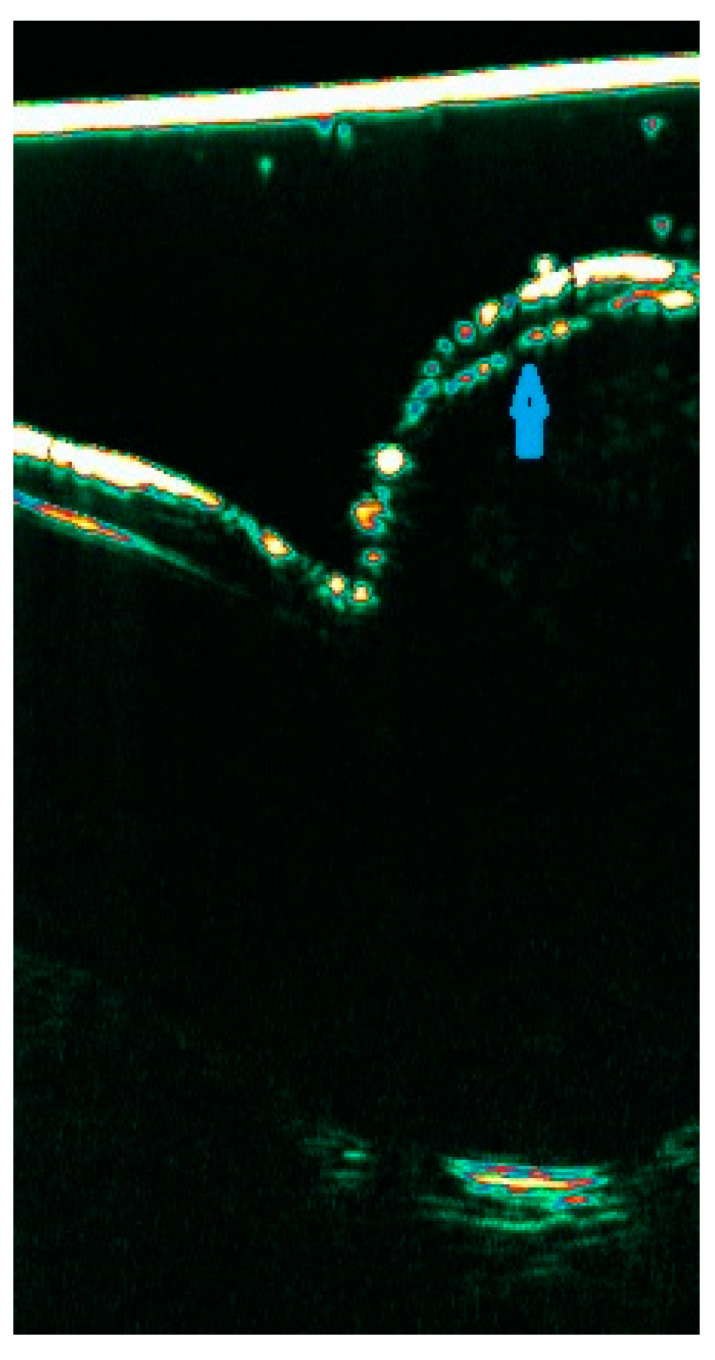
Mucous cyst (marked with blue arrow) presented as a hypoechogenic structure with sharply dermacated hyperechogenic borders in the proximal nail fold (20 MHz).

**Table 1 diagnostics-15-00332-t001:** The sonographic features of different types of alopecia.

	Androgenetic Alopecia	Alopecia Areata	Dissecting Cellulitis of the Scalp	Lichen Planopilaris/Frontal Fibrosing Alopecia	Discoid Lupus Erythematosus
**Hair follicles**	early stage: similar to healthy scalp more advanced cases; reduction in the number of hair follicles and their diversity in terms of width and the location of the distal end in the skin	dilated, inflamed hair follicles with distinct distal ends and a ater-drop-like morphology (Figure 2)	widened hair follicles connected with hypoechoic fluid collections located in the deeper parts of the skin	enlarged, inflamed hair follicles with distinct distal segments resembling cigar (Figure 3)	dilated hair follicles with a tendency to form a homogeneous hypoechoic band below the entrance echo, blurring the boundaries between individual hair follicles
**Other**	NA	NA	NA	a saw-shaped dermal–subdermal border in cases with pronounced scarring	focal doubling of the entrance echo band
**Advanced stages of cicatrical alopecia**	NA	NA	NA	reduction in the number/lack of hair follicles, increased echogenicity of the skin, pronounced dermal/subdermal boundary with the possibility of forming a specific pattern	
**Ref. No**	[18]	[18]	[16]	[19]	[19]

**Table 2 diagnostics-15-00332-t002:** Echogenicity of benign and malignant NU tumors.

Hypoechoic	Hyperechoic	Refrence No
Glomus tumor	Onychopapilloma	[37,40,41,43]
Pyogenic granuloma	Subungual exostosis	[37,39]
Onychomatricoma		[37]
Keratoacanthoma		[37]
Viral wart		[44]
Mucous cyst		[37,42,45]
Squamous cell carcinoma		[37,41]
Melanoma		[37,46]
Subungual hemangioma		[42]

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
