# Peer review of "High-Frequency Ultrasonography in Hair and Nail Disorders—How It May Be Helpful"

_diagnostics, 2025, doi:10.3390/diagnostics15030332_

Round 1
Reviewer 1 Report
Comments and Suggestions for Authors
Nice report on relevant findings where HIFU can be usefull in describing hair and nails. It would be usefull to add an US picture with the normal anatomy of both hair and nail to describe all the different structure in a picture (the manuscript mentions them but they do not correlate to a picture). This will make easier for the practitioner that is not familiarized with these structures to understand it. These are my comments and edits to the manuscript:
- Figure 1: specify which picture belongs to each one of the disorders.
- Table 1: on the early stage of androgenetic alopecia there is an incomplete sentence, please review.
- Table 1: review grammar on the first sentence of LPP/FFP hair follicle, repeat sentence (same sentence twice)
- Table 1: please fix the format (bullets are our of place, lines are missing on some of the sides)
- Table 1: word data on the DLE line. Please delete.
- Line 123: figure 2 is mentioned as part of alopecia areata, but figure 2 correspond to nail unit. Please review and correct it.
- Line 193: please review sentence: "The lower ..... Thumb." Does not contribute anything to the article.
- Figure 3: put markers in the pictures to indicate all the findings you are describing on lines 216-218.
- Figure 4: please put letters on the Figure 4. You specify figure 4b, c and d but there are not letters on the pictures. Also it would be helpful to add markers on the pictures to all the findings you are describing for nail psoriasis.
- Figure 5: please put a marker in the picture that labels the oil spot
- Figure 6: put markers in the pictures to indicate all the findings you are describing on LP.
- Figure 6 label: please correct the spelling of "Lichan Plasu" to lichen planus
- Table 2: please put subungual hematoma in the correct column.
Author Response
Dear Reviewer,
Thank you for giving us the opportunity to submit a revised draft of the manuscript “High-frequency ultrasonography in hair and nail disorders - how it may be helpful" for publication in the special issue of Diagnostics. We appreciate the time and effort that you and the reviewers dedicated to providing feedback on our manuscript and are grateful for the insightful comments on and valuable improvements to our paper. We have incorporated most of the suggestions made by the reviewers. Those changes are highlighted within the manuscript. Please see below, for a point-by-point response to the reviewers’ comments and concerns.
Reviewer 1.
Nice report on relevant findings where HIFU can be usefull in describing hair and nails. It would be useful to add an US picture with the normal anatomy of both hair and nail to describe all the different structure in a picture (the manuscript mentions them but they do not correlate to a picture). This will make easier for the practitioner that is not familiarized with these structures to understand it.
- Comment 1: According to the suggestions, all photos were corrected and described, with the appropriate structures or changes marked on the photos.
These are my comments and edits to the manuscript:
Figure 1: specify which picture belongs to each one of the disorders.
- Comment 2. The answer like in comment 1.
Table 1: on the early stage of androgenetic alopecia there is an incomplete sentence, please review.
- Comment 3: The incomplete sentence was corrected in Table 1. Additionaly the Table 1 was edited as proposed in new version of the article.
- Table 1: review grammar on the first sentence of LPP/FFP hair follicle, repeat sentence (same sentence twice)
- Comment 4: The grammar was corrected in Table 1.
- Table 1: please fix the format (bullets are our of place, lines are missing on some of the sides)
- Comment 5: The format was modified as suggested to the format respected by the Diagnostics template.
- Table 1: word data on the DLE line. Please delete.
- Comment 6: the word was corrected.
- Line 123: figure 2 is mentioned as part of alopecia areata, but figure 2 correspond to nail unit. Please review and correct it.
- Comment 7: The numbers of the figures were reviewed and corrected. Additionally, to be more precise in description in alopecias I prepared separate photos to presented entity ( not in one, as previously). I also added one more photo on mucoid cysts (Fig. 9).
- Line 193: please review sentence: "The lower ..... Thumb." Does not contribute anything to the article.
- Comment 8: The sentence was rewritten to be understood.
- Figure 3: put markers in the pictures to indicate all the findings you are describing on lines 216-218.
- Comment 9: The markers to pictures were added.
- Figure 4: please put letters on the Figure 4. You specify figure 4b, c and d but there are not letters on the pictures. Also it would be helpful to add markers on the pictures to all the findings you are describing for nail psoriasis.
- Comment 10: The markers and descriptions were added.
- Figure 5: please put a marker in the picture that labels the oil spot
Comment 11: The markers were added.
- Figure 6: put markers in the pictures to indicate all the findings you are describing on LP.
Comment 12: The markers were added.
- Figure 6 label: please correct the spelling of "Lichan Plasu" to lichen planus
- Comment 13: the sentence was corrected.
- Table 2: please put subungual hematoma in the correct column.
Comment 14: The subungual hematoma may present as both hypo- and hyperechogenic area- that is why it was in the middle. I reviewed once again the Table 2- and did my best to adjust it to the proposed format.
I hope that the improved version with the supplemented photo descriptions will now be more accessible to the reader and will satisfy Reviewer. I will continue to modify the photos as necessary.
Best regards,
Adriana Polańska
Reviewer 2 Report
Comments and Suggestions for Authors
The main problem of this review is the low quality of the images. I suggest to add in the legend the sonographer and to turn the images so to have epidermis at the top of the image.
Furthermore, hidradenitis that is one the most important skin disease of hair follicle has been omitted.
More references are requested (I suggest to put them in table 1).
Minor issues:
- in the abstract NU is for nail unit
- please remove "epidermis products"
- line 24, the subject of the sentence is missing
- Legend of figure 1. please explain for non-experts which are the US characteristics visible in the images
- Table 1: correct "alopecia"
- Correct legend of figure 2
- correct all the typos...!
Author Response
Dear Reviewer,
Thank you for giving us the opportunity to submit a revised draft of the manuscript “High-frequency ultrasonography in hair and nail disorders - how it may be helpful" for publication in the special issue of Diagnostics. We appreciate the time and effort that you and the reviewers dedicated to providing feedback on our manuscript and are grateful for the insightful comments on and valuable improvements to our paper. We have incorporated most of the suggestions made by the reviewers. Those changes are highlighted within the manuscript. Please see below, for a point-by-point response to the reviewers’ comments and concerns.
Reviewer 2.
The main problem of this review is the low quality of the images. I suggest to add in the legend the sonographer and to turn the images so to have epidermis at the top of the image.
Furthermore, hidradenitis that is one the most important skin disease of hair follicle has been omitted.
More references are requested (I suggest to put them in table 1).
Comment: We prepared the photos according to our knowledge and made every effort to improve the quality of the photos. We replaced the first photo with another photo with fewer artifacts that make it difficult to observe hair follicles. Additionally, we included each of the hair diseases as a separate Figure. In accordance with the suggestions, we described the photos and changed their arrangement. The aim of the work was to present the possibility of observing changes in the hair follicles of the scalp, which is why we did not discuss HS ( to be more precise I added "scalp" to the abstract). The data presented are unique and many of the works summarized in this review are pioneering, hence the scarcity of references. If there are any bibliographic data not included, I kindly ask for help and I will immediately supplement the article in this area. In accordance with the suggestions, the references were added to Tables 1 and also Table 2, consistently.
Minor issues:
- in the abstract NU is for nail unit
Comment: Abbreviation was included to the abstract.
- please remove "epidermis products"
Comment: The epidermis products – this part was removed.
- line 24, the subject of the sentence is missing
Comment: The sentence was corrected.
- Legend of figure 1. please explain for non-experts which are the US characteristics visible in the images
Comment: The all changes in HFUS in scalp and nail diseases were explained .
- Table 1: correct "alopecia"
Comment: It was corrected.
- Correct legend of figure 2
Comment: It was corrected.
- correct all the typos...!
Comment: We corrected all the typos.
I hope that the improved version with the supplemented photo descriptions will now be more accessible to the reader and will satisfy Reviewer. I will continue to modify the photos as necessary.
Best regards,
Adriana Polańska
Round 2
Reviewer 1 Report
Comments and Suggestions for Authors
Thank you for adding all the edits. Please add the following corrections:
1. Table 1 format is not organized. The lines for LPP and DLE are blurred on the advanced stages of alopecia.
2. In the Figure 9, please review the legend. It does not make sense. I suggest to add structure after the word hypoechogenic and delete the word are.
Author Response
Dear Reviewer,
Thank you for giving us the opportunity to submit a revised draft of the manuscript “High-frequency ultrasonography in hair and nail disorders - how it may be helpful" for publication in the special issue of Diagnostics. We appreciate the time and effort that you and the reviewers dedicated to providing feedback on our manuscript and are grateful for the insightful comments on and valuable improvements to our paper. We have incorporated most of the suggestions made by the reviewers. Those changes are highlighted within the manuscript. Please see below, for a point-by-point response to the reviewers’ comments and concerns.
Thank you for adding all the edits. Please add the following corrections:
- Table 1 format is not organized. The lines for LPP and DLE are blurred on the advanced stages of alopecia.
- In the Figure 9, please review the legend. It does not make sense. I suggest to add structure after the word hypoechogenic and delete the word are.
Thank you very much for all your comments.
- In accordance with your suggestions, I have modified Table 1 - although this is the format given by the journal and the divisions between the parts of the table are invisible, making the text difficult to read. I agree with this, so I have changed the orientation in the Table - will this form be more legible?
- I rewrote the legend under the Figure 9.
Best regards,
Adriana Polańska
Reviewer 2 Report
Comments and Suggestions for Authors
Thank you for your kind answer. I noticed a little improvement of the quality of the photos, although Figure 6 again should be turned right.
I see again some typos, i.e. reference in Table 1 and 2.
The paper improved with the tables and references of figures.
I suggest to add more explanations and maybe figures about retronychia and onychopapilloma (Benzecry V, et al. Retronychia as a cause of opportunistic nail infections. Int J Dermatol. 2022)
Author Response
Dear Reviewer,
Thank you for giving us the opportunity to submit a revised draft of the manuscript “High-frequency ultrasonography in hair and nail disorders - how it may be helpful" for publication in the special issue of Diagnostics. We appreciate the time and effort that you and the reviewers dedicated to providing feedback on our manuscript and are grateful for the insightful comments on and valuable improvements to our paper. We have incorporated most of the suggestions made by the reviewers. Those changes are highlighted within the manuscript. Please see below, for a point-by-point response to the reviewers’ comments and concerns.
Reviewer 2.
Thank you for your kind answer. I noticed a little improvement of the quality of the photos, although Figure 6 again should be turned right.
I see again some typos, i.e. reference in Table 1 and 2.
The paper improved with the tables and references of figures.
I suggest to add more explanations and maybe figures about retronychia and onychopapilloma (Benzecry V, et al. Retronychia as a cause of opportunistic nail infections. Int J Dermatol. 2022)
Response: Thank you very much for all your comments. In Figure 6, I changed the orientation as suggested. The typos have been corrected, I have also modified Table 1 to make it more readable. As mentioned, I have added the suggested reference. All features of retronychia have been included in this article.
Best regards,
Adriana Polańska